# Ecological and Biotechnological Relevance of Mediterranean Hydrothermal Vent Systems

Carmen Rizzo [1,2,†], Erika Arcadi [3,†], Rosario Calogero [3], Valentina Sciutteri [3], Pierpaolo Consoli [3,*], Valentina Esposito [4,5], Simonepietro Canese [5], Franco Andaloro [6] and Teresa Romeo [7,8]

1.  Stazione Zoologica Anton Dohrn, Ecosustainable Marine Biotechnology Department, Sicily Marine Centre, Villa Pace, Contrada Porticatello 29, 98167 Messina, Italy; carmen.rizzo@szn.it
2.  Institute of Polar Sciences, National Research Council, Spianata S. Raineri 86, 98122 Messina, Italy
3.  Stazione Zoologica Anton Dohrn, Integrative Marine Ecology Department, Sicily Marine Centre, Villa Pace, Contrada Porticatello 29, 98167 Messina, Italy; erika.arcadi@szn.it (E.A.); rosario.calogero@szn.it (R.C.); valentina.sciutteri@szn.it (V.S.)
4.  National Institute of Oceanography and Applied Geophysics, Via Auguste Piccard 54, 34151 Trieste, Italy; vesposito@inogs.it
5.  Stazione Zoologica Anton Dohrn, Research Infrastructures for Marine Biological Resources Department, Via Po 25, 00198 Roma, Italy; simonepietro.canese@szn.it
6.  Stazione Zoologica Anton Dohrn, Integrative Marine Ecology Department, Sicily Marine Centre, Lungomare Cristoforo Colombo n. 4521, Località Addaura, 90149 Palermo, Italy; franco.andaloro@szn.it
7.  Stazione Zoologica Anton Dohrn, Integrative Marine Ecology Department, Sicily Marine Centre, Via dei Mille, 46, 98057 Milazzo, Italy; teresa.romeo@szn.it or teresa.romeo@isprambiente.it
8.  National Institute for Environmental Protection and Research, Via dei Mille 46, 98057 Milazzo, Italy
*   Correspondence: pierpaolo.consoli@szn.it
†   These authors contributed equally to this work.

**Abstract:** Marine hydrothermal systems are a special kind of extreme environments associated with submarine volcanic activity and characterized by harsh chemo-physical conditions, in terms of hot temperature, high concentrations of $CO_2$ and $H_2S$, and low pH. Such conditions strongly impact the living organisms, which have to develop adaptation strategies to survive. Hydrothermal systems have attracted the interest of researchers due to their enormous ecological and biotechnological relevance. From ecological perspective, these acidified habitats are useful natural laboratories to predict the effects of global environmental changes, such as ocean acidification at ecosystem level, through the observation of the marine organism responses to environmental extremes. In addition, hydrothermal vents are known as optimal sources for isolation of thermophilic and hyperthermophilic microbes, with biotechnological potential. This double aspect is the focus of this review, which aims at providing a picture of the ecological features of the main Mediterranean hydrothermal vents. The physiological responses, abundance, and distribution of biotic components are elucidated, by focusing on the necto-benthic fauna and prokaryotic communities recognized to possess pivotal role in the marine ecosystem dynamics and as indicator species. The scientific interest in hydrothermal vents will be also reviewed by pointing out their relevance as source of bioactive molecules.

**Keywords:** hydrothermal vents; benthos; prokaryotic; environmental change; bioactive molecules

## 1. Introduction

The extreme conditions have characterized our planet since primordial eras and have been maintained over time in some specific sites with unique features, ranging from extremely cold and hot regions, anoxic basins, deep sea, hydrothermal vents, and areas with harsh conditions of pressure, salinity, nutrient concentration, and oxygen availability. All these conditions stimulate a constant questioning about the limits of life. The fascination that these environments exert on the research world is polyvalent, since a deeper knowledge

of them allows an understanding of the past as well as a look to the future, as an important resource for predictive studies on future scenarios and for the possibility of new discoveries.

Among extreme environments, hydrothermal vent systems (HVSs) are special ecosystems located in tectonically active settings, commonly classified as terrestrial, deep-sea (DHV), and shallow-sea (SHV), depending on location and water depth [1]. Due to the lesser accessibility compared to their terrestrial counterpart, marine hydrothermal systems (HSs) have been less frequently investigated, despite being widespread, and new ones are frequently discovered. Although SHVs and DHVs differ for a cut of just 200 m, they have many ecological differences. SHVs occur in a number of highly diversified active settings, including submarine volcanoes, island and intra-oceanic arcs, ridge environments, intraplate oceanic volcanoes, continental margins, and rift basins [1]. The source of water is suggested to be a mixture of meteoric, magmatic, groundwater, and/or seawater, so they could be considered intermediate between terrestrial and off-shore geologic environments. For these reasons, due also to their location in the euphotic zone, SHVs are very complex and dynamic. Differently, DHVs are characterized by a complete absence of light, extreme temperature (up to 400 °C) and pressure, and strong thermal gradients between vent fluids and the surrounding seawater [2].

HSs in the past were considered prohibitive to any form of life, but the research carried out in recent years has shown that at the base of these systems, there is an intense microbial activity capable of sustaining an active trophic web, despite the unfavorable environmental conditions. Hydrothermal conditions have a low impact on microorganisms, which tolerate strong physical and chemical gradients, thanks to nutritional requirements and overall metabolic pathways ideally suited to such ecosystems [3]. In both types of environments (DHVs and SHVs), microbial communities play a crucial ecological supporting role, by transforming into biomass the inorganic compounds released from vent emissions. Thus, microorganisms are involved in processes such as organic matter synthesis, breakdown or mineralization, metal cycles, and interactions with fauna (Figure 1) [4]. Unlike DHVs, where the ecosystem functioning is mainly based on chemosynthesis, the primary production at SHVs is based both on photosynthesis and chemosynthesis processes, granted by sunlight and oxidation of reduced compounds as energy sources [5–7].

These microbial processes are at the base of the hydrothermal trophic food chain and involve several forms, i.e., free-living microorganisms associated with the dismissed vent fluids, free-living microorganisms on the surface of flowing vent waters, or symbiotic forms associated with invertebrates [8]. Indeed, the occurrence and distribution of benthic organisms in DHV is controlled by the outflowing fluids enriched in iron, sulfides, and gases, utilized by the symbiotrophic invertebrate [9–12]; in SHV, it is affected by the distribution of the free-living microbial communities [13].

The surprising discovery of life in these seemingly inhospitable environments has fueled the growing interest of researchers, but the challenge goes far beyond studying the diversity they harbor. This review aims at focusing on the biodiversity of marine hydrothermal vents of the Mediterranean area, with a special hotspot on necto-benthic and microbial communities. Moreover, the scientific relevance of such environments as natural laboratories and ecological models to study future scenarios, to understand how their conditions have shaped the living communities in the past, and as valuable source in the bioprospecting field, will be explored. Definitely, it is aimed at highlighting the main gaps currently existing for these still unknown environments and their extraordinary biotechnological potential.

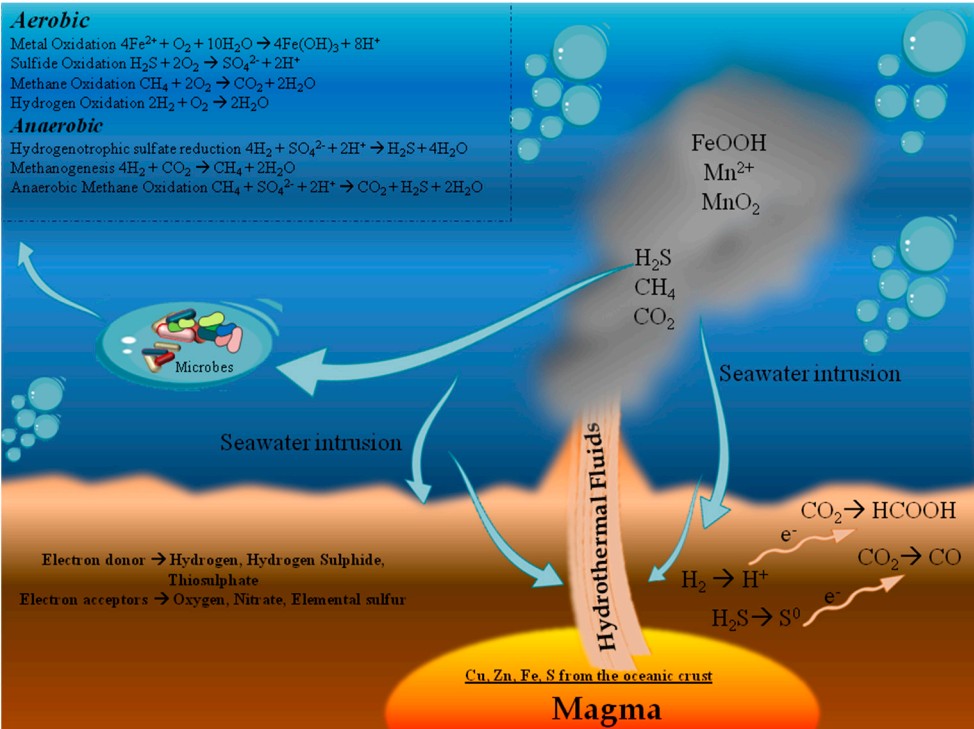

**Figure 1.** Graphic showing a vent forming a hydrothermal plume with interactions between seawater, magmatic fluid and crustal rocks. Release of some chemicals (i.e., Manganese, iron, iron oxyhydroxide, carbon dioxide, hydrogen sulfide, methane) is generated by the mixing of the hot hydrothermal fluids and cold seawater. Simplified view of microbial metabolisms in HVSs. The characteristics are generalized and do not refer to a particular vent location.

## 1.1. Environmental Parameters

In the past, we referred to HSs as environments that were not easily accessible, poorly studied, characterized by the paucity of life-forms, and, above all, as places located at very high depths. Over the years, the discovery of SHVs demonstrated that hydrothermal sites can be found also at shallow depths. Conventionally SHVs are systems found up to 212 m deep, while DHVs are those located beyond 212 m depth [13]. The two main environmental differences between SHVs and DHVs are, thus, represented by temperature and depth, which affect light radiation and pressure. Moreover, the two HSs could differ for the presence/absence or the different concentration of reduced compounds and metals.

### 1.1.1. Temperature

Temperature strongly affects the distribution of the biota, since only few eukaryotes are well adapted to high temperatures. It controls the distribution of phototrophic organisms in both oxy and anoxic conditions [14]. In SHVs, the temperature of the fluids ranges between 10 and 119 °C and it can increase up to 95.8 °C in the sediments [15–19]. In DHVs, the temperature of the fluids can exceed 400 °C, and seawater penetrating deep inside the earth's crust can create fluids that escape at a temperature of 1200 °C [20]. Venting fluids are usually classified into low (up to ~50 °C), medium (up to ~200 °C), and high temperature (up to 400 °C and more) [21]. Despite the occurrence of these extreme conditions, it has been found that most of the biota associated with vents live at temperatures between 10 and 25 °C [22].

Mediterranean HVs are equally represented by shallow sites and deep sites, with temperature exceeding the boiling point of water (see Vulcano and Panarea Islands), but never reaching values above 135 °C, probably due to their location within 1200 m.

### 1.1.2. Light radiation

Unlike DHVs, where primary production is based on chemosynthesis exclusively, in SHVs, it is sustained also by photosynthesis thanks to the availability of light radiation [13,14,23–26]. The coexistence of these two metabolisms, the high concentrations of $CO_2$, and the easier accessibility make the SHVs model sites to study global environmental changes, such as ocean warming and ocean acidification [5,27].

### 1.1.3. pH

Hydrothermal fluids in SHVs are usually rich in $CO_2$ (95–98% of the total fluids composition), present in the form of gas bubbling, with negligible coemission of sulfide and methane. The active emissions of these $CO_2$-dominated fluids alter the seawater carbonate chemistry around the HVs resulting in pH values lower than the ambient seawater (i.e., acidification), with effects on the whole ecosystem.

Moreover, the mineralogical diversity and complexity of HVs is due to a wide variety of inorganic chemical compounds released by the hydrothermal activity. Such minerals could act as inorganic surfaces to favor the formation of organic molecules, but most importantly, they could generate chemical gradients stimulating the interaction between electron donors and acceptors.

The formation of local pH gradients makes HVs natural laboratories for assessing the impacts of ocean acidification on the structure of marine ecosystems because they expose entire communities to a lifetime of elevated $CO_2$ levels [10].

### 1.1.4. Chemical Elements

The difference in chemical concentrations is determined by the fluids' composition. In general, DHVs fluids are characterized by high concentration of reduced substances and metals, gas phase present only in dissolved form due to the pressure, and extremely high values of $H_2$, $H_2S$, and $H_4$ [13].

However, it has been observed that when there is a low concentration of Cl, fluids are more enriched in $H_2$ and $CH_4$, while a higher concentration of metals occur in the fluids rich in Cl [28–30]. Generally, the underwater hydrothermal environments are subjected to mineral deposits events, including also heavy metal deposition. This natural phenomenon is due to the violent volcanic eruptive manifestations bringing incandescent scoria out of the crater. These scoria, which come out at high temperatures, precipitate when in contact with the sea water, leading to the accumulation of various elements [31], which are not bioavailable nor soluble [32].

Differently from DHVs, in SHVs, a substantial role in the formation of vent fluids is played by meteoric water, gases are present in their free form, and the water column is oxygenated [13]. This implies that the gaseous fluids are visually perceived with the formation of free gas bubbles, causing an alteration in the chemistry of the fluids, from the gas to the aqueous phase [33]. Hydrothermal fluids mixed with oxygen-rich water determine the formation of redox microgradients, with well-defined biological communities [13]. The differences related to chemical concentrations could be also due to terrigenous inputs, which are more consistent in SHVs compared to the deep ones and represent a significant source for the carbon cycle and primary production [34].

As regards fluid composition, high concentrations of trace elements have been also reported in Mediterranean SHVs [31,35,36]. Moreover, studies conducted by Handley et al. [37] and Price et al. [24,25] on the Aegean arc suggested that the oxidation of arsenite and the arsenate reduction are the thermodynamically favorable metabolic pathways [34,38,39].

Other physico-chemical parameters have received less attention compared to those described above; however, some profiles seem rather similar across different HVs sites. For instance, salinity does not show a significant differences compared to the ambient seawater values [27,40]; oxygen concentration along with redox potential is usually lower at vents and reflect oxygen consumption due to the oxidation of metals, such as iron, and reduced gases, such as $H_2$ and $H_2S$ [40,41]; high total organic carbon (TOC) concentration as well

as Biopolymeric Organic Carbon (BPC) increase in the sediment at vent sites [42,43], with protein dominating the carbon pool, followed by lipids and carbohydrates. In contrast, phytopigments showed an opposite trend with increasing values at increasing distances from the vent openings [42,43]. Interestingly, dissolved nutrient concentration (e.g., $SO_4^{2-}$, $NO_3^-$, $NO_2$, and $SiO_2$) show site-specific profiles [27].

### 1.2. Hydrothermal Systems in the Mediterranean Sea

According to the InterRidge Vents Database (InterRidge Vents Database, Ver. 3.4, Accessed 18 March 2021, http://vents-data.interridge.org/) hydrothermal sites are evenly distributed in the Mediterranean Sea with 10 SHV complexes and 13 DHV, listed in Table 1.

The area of central Tyrrhenian Back-Arc Basin includes the SHVs of Ischia Island (Castello Aragonese), of Naples Gulf (Capo Miseno and Phlegrean Fields), and the deep Marsili vulcanic complex. In the southern Tyrrhenian Sea, the Aeolian Arc is divided into: SHVs of Panarea, Vulcano, and Palinuro vulcanic complex; DHVs with Palinuro seamount, Vavilov, Enarete, Eolo, Sisifo, Secca del Capo, and Stromboli; the Hellenic Arc located in the Aegean Sea comprises the SHVs of Kos, Yali, Methana, Nisiros, Milos, and the deep Santorini, Kolumbo; finally, the Empedocle Banks with the DHVs of Graham, Terrible, and Nerita Banks are located off Pantelleria Island in the Strait of Sicily (Figure 2).

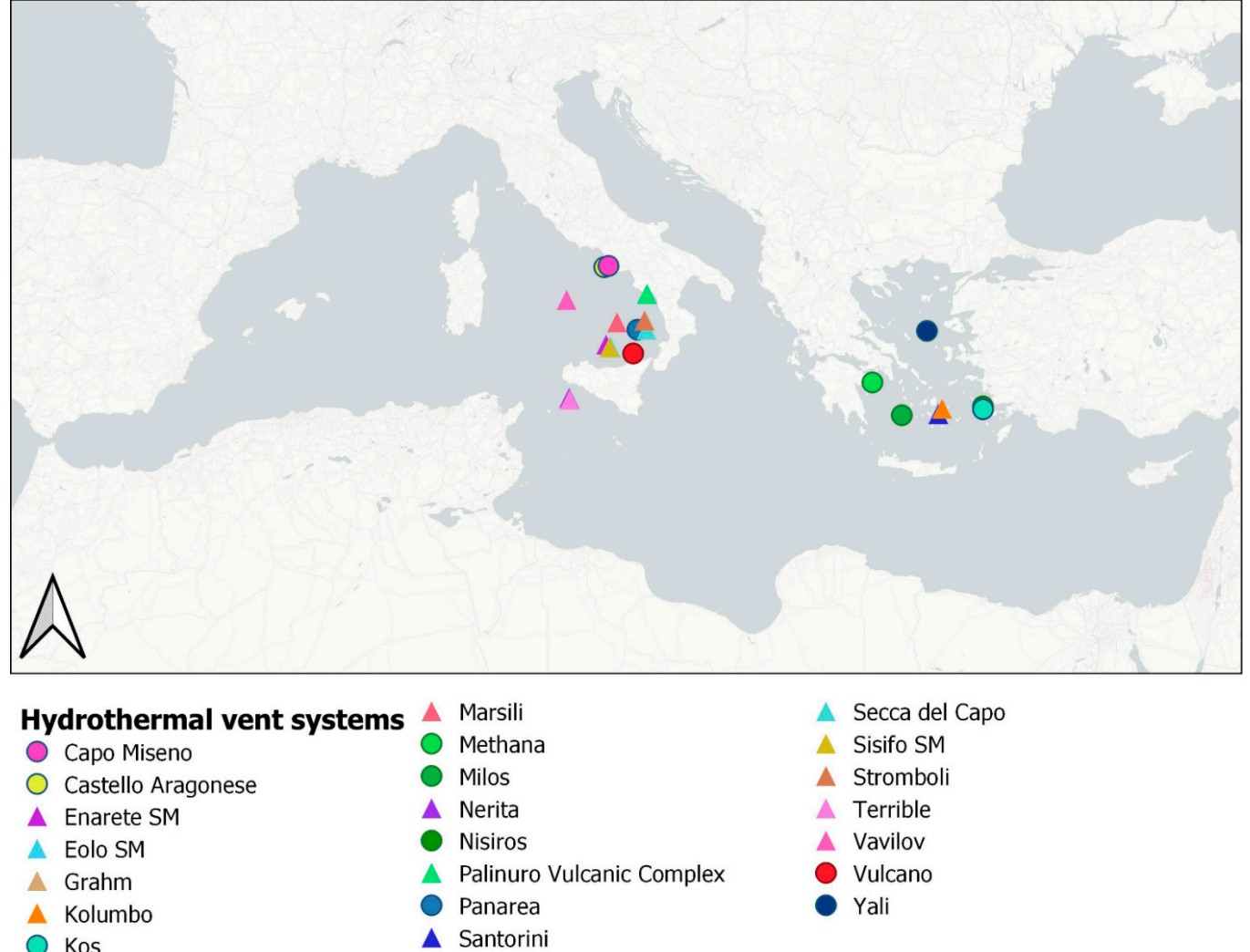

**Figure 2.** Main hydrothermal vent systems in the Mediterranean area: central Tyrrhenian Back-Arc, Aeolian Arc, Aegean Arc and strait of Sicily. Triangles are deep site, while rings are shallow site.

Mediterranean hydrothermal fields have been reported in various areas, mainly on arc volcanoes, back-arc basins, and hot spot volcanoes. The first shallow emissions were identified in the Mediterranean Sea via scuba, with evident underwater fumarolic sulfide deposits detected at the Levante Bay in Vulcano Island (Aeolian Arc) [44] and in the areas of Nisiros, Methana, and Kos (Hellenic Arc) [45]. Among the first DHVs identified in the Mediterranean Sea were Stromboli [46] and Palinuro Seamount [47]. The discovery of all the other Mediterranean hydrothermal vents occurred after the 1990s. In the last five years, a seabed exploration based on multibeam echo-sounder data conducted by Coltelli et al. [48] recognized the small submarine volcanic field Graham in the surroundings of the Graham Bank (northwestern portion of Sicily Strait, Italy), subsequently to the 1831 submarine eruption which originated the Ferdinandea Island. Three banks were discovered: Nerita, Terrible, and Graham, to date defined only at geomorphological level.

The Mediterranean hydrothermal complexes show environmental parameters that vary widely among different systems as well as from those found in other geographical areas. They are characterized by a range of 0–1200 m depth, with temperatures ranging between 13–135 °C. Some sites can receive a larger amount of terrigenous sediments as they are located more closely to the land, but they all share the presence of a well-noted gaseous composition ($CO_2$, $H_2S$, $CH_4$). Not all Mediterranean vents are characterized by hot emissions, as revealed by the low temperature value (between 13–25 °C) detected in the deep Ischia Island, Palinuro Cape, some areas of the Panarea Volcanic Complex, Yali, Methana, Nisiros, Santorini, and Kos HVs.

The Aeolian archipelago is one of the most known Mediterranean volcanic arcs, located in the southern Tyrrhenian Sea in correspondence of a continental crust (thickness 15–20 km) composed of seven islands and nine sea mountains with an extension of 22 km². Each island is characterized by a different tectonic and magmatic history [49]. Calanchi et al. [50]) distinguished the western islands (Alicudi, Filicudi, and Salina), with a calcalkaline feature, from the central eastern islands (Vulcano, Lipari, Panarea, and Stromboli), characterized by high-potassium calcalkaline (HKCA), shoshonitic, and alkaline potassic rocks. In this last area, the high-pressure gas has created small craters and exhaling sites, as in the case of the Bottaro Islet (Panarea island, temperature 30 °C, presence of colloidal material containing sulfur salts), the so-called 'Sink-Hole' (an exhaling area with fluid temperature of 90 °C) and the 'black smoker' (single emission point with sulfur-rich emissions and fluid temperature of 110 °C) [51].

To date, the Panarea hydrothermal complex represents one of the most active HSs in the Mediterranean [38]. Other than the main island Panarea, it includes the islet of Basiluzzo and the eastern archipelago of Dattilo, Panarelli, Lisca Bianca, Bottaro, Lisca Nera, and Formiche islets [52]. A heterogeneous magma source was suggested for this area, with the coexistence of andesitic and dacitic lava domes, flows, pyroclastics, and calcalkaline and shoshonitic rocks [53]. Panarea is the smallest island of the Aeolian Arc, with a total elevation of 1600 m above the seafloor [52] and hosts hydrothermal mineralization processes based on discharge of low pH fluids with temperatures around 300 °C [54]. Thanks to the easy access of the sites, it was possible to characterize the area well before and after the great degassing activity occurred in November 2002 [33,52,55]. Following the violent release of gas in the area overlooking the islets east of Panarea, numerous experiments have been conducted and are still underway in order to understand the origin, nature, and effects of these emissions. Recently, some environmental parameters (temperature, redox potential, pH, and oxygen concentration) have shown different profiles compared to those previously reported [56]. Indeed, previous studies [57] reported extreme temperature values (>50 °C) and acid pH conditions from Panarea vents, which, from more recent monitoring activities, seem to be attenuated (data not shown). Given its extreme and sudden variability, the area is continuously monitored [58], also to reveal consequent changes in biological communities. Several studies have shown that $CO_2$ emissions in the Panarea area are quite evident and changeable, and affect micro- and macrofauna biodiversity [38,59–61]. The growing interest and consequently the assiduous investigation of the Panarea seabed between 2013 and 2015

allowed to discover a new HS located at the base of the escarpment between Panarea and Basiluzzo called "Smoking land", characterized by the presence of an extensive and active field of hydrothermal chimneys [39].

Vulcano is the southernmost island of the Aeolian Arc, presenting high-temperature fumaroles near La Fossa crater with temperature gases ranging from 100 to 450 °C (reported by Amend et al. [62]). On the eastern coast of the island, emissions rich in $CO_2$ and temperatures varying from ambient to about 100 °C were detected around Levante Bay. The La Fossa crater fumaroles are mainly constituted by high concentration of $CO_2$ and considerable amounts of HCl, $SO_2$, $H_2S$, HF, and CO [63], while the emissions in Levante Bay are mainly representative of hydrothermal fluids, with higher concentration of $CO_2$, $CH_4$, and $H_2S$ and a lower concentration of CO [62]. In this area, a gradient in seawater carbonate chemistry runs parallel to the coast for about 400 m toward north and makes the site suitable for studies on ocean acidification [41]. A geochemical similarity between the Aeolian eastern islands and the volcano Vesuvius and Phlegrean Fields was reported by Peccerillo et al. [61].

However, a 90 km volcanic chain lies in the southern Tyrrenian Sea. Here, the largest European submarine volcano is located, renowned for its precious mineral deposits. The Palinuro volcanic complex, consisting of five coalesced volcanic edifices, is located at the end of the Sangineto line, a fault system which separates Calabria and southern Appennines [64]. The depth of the complex varies from a minimum of 70 m in the east to a maximum of 570 m in the west. The volcanic rocks have revealed a calc-alkaline affinity basalt/basaltic andesite composition [65–67]. Here, sediments are rich in sulfides, manganese, iron oxide deposits, and clay minerals [53]. The most recent geological studies on Palinuro seamount were conducted by Cocchi et al. [68], while the only published data on the microbial communities of this site are by Mattison et al. [69]. To the best of our knowledge, these sites remain still scarcely explored.

The Ischia Island and The Phlaegrean Fields constitute the volcanic district of the west Bay of Naples, dating back to the Plio-Quaternary. The HS of Ischia is located in the northwest of the area, it extends in both shallow and deep waters [70,71], and is characterized by important events of secondary volcanism. Indeed, the area presents submerged volcanic and monogenic edifices around the island and the nearby continental shelf. The geomorphology of the area is very complex [72], and the site is characterized by a release of bubble plumes rich in 90–95% $CO_2$, 3–6% $N_2$, 0.6–0.8% $O_2$, and 0.08–0.1% $CH_4$ (no sulfur detected) [73] and mean seawater pH values were between 8.14 and 6.57 [10]. As reported by various authors [74,75], the seawater near the rocky substrate is strongly acidified, with $pH_T$ in the range of 6.6–7.2. However, one of the most exploited areas for research on the impact of ocean acidification is the shallow HS of the Castello Aragonese, where low temperature $CO_2$ emissions give rise to horizontal pH gradients ranging from 6.6 to 8.12 and extended to approximately 150 m of the coastline on the northern and southern sides of the Castello Aragonese [75]. This HS is the remnant of an old volcano dating back to 150,000 to 75,000 years ago [76] and is located in the proximity of a fault corresponding to a fracture present in the complex of Ischia Island.

The HSs located in the Aegean Sea represent one of the most seismically active regions on earth [77,78]. The Hellenic Volcanic Arc is a 5 Ma old volcanic complex, formed in the pre-Alpine to Quaternary continental crust of the Hellenic Subduction System as a consequence of the interaction between the northward of the last remnant of the oceanic crust of the African plate and the southern edge of the active margin of the European plate [79,80]. Milos, located in the middle of the volcanic complex, is the most studied hydrothermal site in the Aegean, as witnessed by the numerous micro-sites recently subjected to observation, namely Palaeochori Bay on the southeastern coast of Milos, several sites in Voudia Bay in the northeastern area of Milos, Adamas Bay in the central zone, Paradise Beach and Brostherma in Kos, and Methana on the southern coast. The sediment temperature in these areas can achieve 85 °C. Most studies in this area are focused on the study of benthic

communities in relation to the hydrothermal activity [81–83], and some reports about microbial communities have been also provided [43,84].

To date, we still continue to scour the bottom of the Mediterranean Sea, like the oceans of the whole world, exploring the unknown underwater world.

**Table 1.** List of hydrothermal vent systems in the Mediterranean area.

| Site | Vent | Max Temperature (°C) | Max Depth (m) | Discovery | Coordinates | | References |
|---|---|---|---|---|---|---|---|
| | | | | Shallow Mediterranean Hydrothermal vents | | | |
| Aeolian Arc Panarea | Smoking Land Black Point Hot/Cold La Calcara Bottaro Crater | 135 | 200 | 1991, SCUBA | 38°63′ | 15°06′ | [33,38,39,51–55] |
| Aeolian Arc Vulcano | Levante Bay | 120 | 8 | 1969, SCUBA | 38°24′ | 14°57′ | [14,85] |
| Aeolian Arc Palinuro Vulcanic Complex | Grotta Azzurra, Grotta Sulfurea | 25 | 30 | 1990, SCUBA | 40°01′ | 15°26′ | [47,64,65] |
| Ischia Island | Castello Aragonese | ND * | 10 | 1954, from the beach | 40°44′ | 13°56′ | [10,70,71,75,76,86] |
| Naples Gulf | Capo Miseno Phlegrean Fileds | ND * | 20 | 1991, SCUBA | 40°46′ | 14°5′ | [14,61,85] |
| Hellenic Arc Milos | Milos | ND * | 200 | 1987, SCUBA 1992 Minirover ROV deeper sites | 36°41′ | 24°23′ | [45,46,76–80,86] |
| Hellenic Arc Kos | Brostherma Kephalos Bay Paradise Beach | ND * | 10 | 1983, SCUBA | 36°51′ | 27°14′ | [45,76–80,86] |
| Hellenic Arc Yali | Yali | ND * | 3 | 1990, SCUBA | 39°1′ | 25°16′ | [76–80,86] |
| Hellenic Arc Methana | Methana | ND * | 10 | 1997, SCUBA | 37°36′ | 23°21′ | [45,76–80,86] |
| Hellenic Arc Nisiros | Lefkos Bay | ND * | - | 1994, SCUBA | 36°56′ | 27°14′ | [76–80,86] |
| | | | | Deep Mediterranean hydrothermal vents | | | |
| Tyrrhenian Back-Arc | Marsili | ND * | 800 | 2006, ROV Cherokee | 39°15′ | 14°23′ | [41,64,67] |
| Aeolian Arc | Enarete SM | ND * | 600 | 2011, ROV Hercules | 38°62′ | 14°0′ | [85,86] |
| | Eolo SM | ND * | 800 | 2011, ROV Hercules | 38°58′ | 14°16′ | [85,86] |
| | Sisifo SM | ND * | 1200 | 2011, ROV Hercules | 38°34′ | 14°7′ | [85,86] |
| | Secca del Capo | ND * | 240 | 1995, ROV Comex | 38°63′ | 14°85′ | [54,86] |
| | Vavilov | ND * | 3500 | 1959, Akademik Valilov | 39°51′ | 12°45′ | [85,86] |
| | Stromboli | ND * | 250 | 2011, ROV Hercules | 38°78′ | 15°21′ | [46,86] |
| Aeolian Arc Palinuro Volcanic Complex | Palinuro SM | ND * | 1000 | 2011, ROV Hercules | 39°55′ | 14°7′ | [41,47,64,67,86] |
| Hellenic Arc Santorini | Santorini | ND * | 400 | 2006, ROV Thetis | 36°43′ | 25°40′ | [76–80,86] |
| Hellenic Arc Kolumbo | Kolumbo | 224 | 500 | 2006, ROV Hercules | 36°52′ | 25°48′ | [76–80,86] |
| Empedocle Banks | Grahm | ND * | 250 | ROV | 37°10′ | 12°42′ | [48] |
| | Nerita | ND * | - | | 37°10′ | 12°43′ | |
| | Terrible | ND * | - | | 37°07′ | 12°43′ | |

ND *, not detected.

### 1.3. Scientific Interest

Several studies on hydrothermal complexes found in the Mediterranean Sea have been focused on the ecological feature description, from micro to macro scale [14,85,86]. Moreover, a number of paper dealt with the distribution and characterization of microbial and benthic communities and their relationship with the surrounding matrices. Numerous microorganisms have been isolated and cultivated from the Aeolian Arc vents, both mesophilic [87–90] and thermophilic [91–94].

The underwater hydrothermal environments have been subjected also to studies on mineral deposits, especially heavy metals. This natural phenomenon foresees that the

violent volcanic eruptive manifestations bring incandescent scoria out of the crater that precipitate in contact with sea-water, leading to the accumulation of various elements [31], which are neither bioavailable nor soluble [32]. The studies available on the Mediterranean vents reflects the fact that temperature is the most shaping factor, and could be divided in those focused on temperate vents (around 13–30 °C) in which the bento-necto-pelagic macrofauna biodiversity diversity is mainly explored, and those focused on the hottest sites, mainly referred to the study of the prokaryotic communities.

As has been pointed out in the previous sections, the scientific interest towards marine HSs is multifaceted. The permanence in these sites of harsh conditions that reflect the remoteness of our planet could be of great interest for international research community engaged in studies on the origins and dynamic of primordial earth, evolution processes, volcanic events, and hydrothermal deposits.

Beyond this, HSs represent a unique opportunity to face the current challenge related to oceanic acidification processes and climate change. Marine environments are strongly threatened by global warming, currently representing one of the most relevant environmental issues, whose future effects on ecosystems and living organisms are mostly unknown. The impacts that will take place as a result of increasing temperature, rising sea levels, and ocean acidification are seriously worrying aspects for the scientific community, which still does not know how to answer numerous questions on the theme [95]. The current rate of emissions in the atmosphere led to estimate an unavoidable increase of $CO_2$ partial pressure at the ocean surface in a short time, with consequent alterations of carbonate chemistry and pH lowering, thus increasing ocean acidification. In fact, we do not know how exactly living organisms will react to these changes, how quickly they will adapt, and which strategies they will adopt eventually, nor which evolutionary processes will occur. In this regard, naturally acidified systems such as $CO_2$ vents can provide insights on the long-term effects of high $pCO_2$ and low pH exposure on marine organisms at the ecosystem level, though at a local scale [96]. Specific investigations were carried out on the effects of $CO_2$ on fish species [97–99]. The most studied area in these regards are Levante Bay (Vulcano island) and Castello Aragonese (Ischia Island), considered useful natural laboratories to assess the effects of ocean acidification, due to their continuous and great natural $CO_2$ emissions acidifying the surrounding seawater [100–102]. Kumar et al. [103] reported the effects of $CO_2$ on *Sargassum* in the area of Castello Aragonese and the influence on its related bioproducts. Similarly, several studies were focused on the effects of hydrothermalism on living communities [59,60] in the area of Panarea and explored the effects of $pCO_2$ variations on seagrass as *Cymodocea nodosa* and *Posidonia oceanica* [100,104] in the Aeolian Archipelago. Recently, Mirasole et al. [97] carried out studies to evaluate the influence of $CO_2$ on the changes in the shape of the otoliths of some coastal fish species of Vulcano Island.

Although Ischia and Vulcano appears as the two most studied sites in recent years from an ecological and biological point of view [105–107], the Ischia vent site remains little explored in relation to the structure of the microbial communities of water and sediment. To the best of our knowledge, the only studies on the microbial communities are referred to bacterial symbionts of ascidians [108] and of corals [109]. To date, most of the investigations in these sites were focused on the impact of global environmental change on eukaryotes, while the susceptibility of microorganisms has been less studied. This represents a very serious gap, considering the pivotal role that microbial communities play in the balances and dynamics of marine ecosystems, as paramount actors in reshaping the oceans and atmosphere, supporting all higher trophic life forms [110].

Last but not least is the significance of marine HSs in the era of biodiscovery. Extreme environments are widely recognized as biodiversity sensing zone, in which adverse environmental conditions determine special adaptation and survival strategies in the associated organisms. One of the priorities in the field of bioprospecting is the discovery of new species or new molecules that have a wide range of applications, and HSs have the perfect requirements to meet these needs. The hottest sites have gained the interest of researchers

due to the physiological adaptation to high temperatures evolved by microorganisms, and for these reasons, the microbiological and molecular approaches have been generally preferred for the study of these areas. In the last few decades, studies in hydrothermal sites showed previously unknown diversity of thermophilic microorganisms [111–113]. As an example, the area of The Phlegrean Fields (Capo Miseno, Tyrrhenian back-arc) is well suited for the research of thermophilic and hyperthermophilic microorganisms. The first studies have been carried out on the microbial community of the sediments located in correspondence with the emissions [36], while the exploration of symbiotic bacteria associated with polychaetes is more recent [4]. Similarly, the Levante Bay in Vulcano, considered one of the hottest sites in the Mediterranean (together with Panarea and Kolumbo), with a temperature of just over 100 °C a few meters from the surface, has been mostly investigated for the search of thermophiles and hyperthermophiles Bacteria and Archaea [114–118] and their bioproducts [118–121].

Several studies have already highlighted the occurrence of extremophilic microorganisms able to produce bioactive molecules, such as exopolysaccharides, extremozymes, and antimicrobial compounds [56,92]. Moreover, the development of new and improved research approaches, as in the case of the omics techniques, could come in to cover the still unknown part of microbial biodiversity, such as the fraction of microorganisms currently uncultivable, therefore revealing their potential.

## 2. Necto-Benthic Communities in Hydrothermal Systems

The knowledge on the necto-benthic communities of hydrothermal vents in the Mediterranean Sea is still fragmented. Biological investigations initially took place in the Aegean Sea (Milos vents) and the South Tyrrhenian Sea (Panarea and Vulcano vents) and were mainly focused on the description of the communities inhabiting these sites [14,85]. Nowadays, the majority of the information available on the organisms living at SHVs derives from ocean acidification vent-based studies, focusing particularly on the effects of elevated $CO_2$ concentrations and low pH on local communities (see Foo et al. [122] and references therein). Mediterranean DHVs have been less explored compared to their shallow counterparts, although the discovery of new sites as well as the occurrence of chemosymbiotic macrofauna are strongly encouraging scientific investigations in this regard [123–125].

### 2.1. Physiological Responses to Environmental Stressors

Warm temperature, gas emissions including $CO_2$ and $H_2S$, low pH, and the input of trace elements are the most shaping factors affecting the physiology of organisms nearby HVs, therefore determining their distribution in and out the vent area. In Palaechori Bay, southeastern coast of Milos, fluctuations in sea water temperatures due to SHVs emissions select algal species with warm-water affinities, suggesting a possible role of shallow water hydrothermal vents as stepping stones for the migration of allochthonous thermophilus species across the Mediterranean Sea [82]. This role of SHVs is also confirmed by the presence of alien macrophytes recorded by Gaglioti and Gambi [126] in one of the highest-temperature $CO_2$ vent systems of Panarea island.

Early observations on Anthozoa assemblages in Palaeochori Bay (Hellenic Arc) also revealed high abundance of several small-sized species, a condition reflecting the so-called "eastern nanism", i.e., the higher occurrence of dwarf specimens in the eastern compared to western Mediterranean Sea [127]. Nonetheless, dwarfism might represent an adaptive strategy for benthic organisms inhabiting SHVs to compensate increased physiological stress and shell dissolution [99,128–131]. More specifically, the gastropods *Cyclope neritea* and *Nassarius corniculus* living at Vulcano SHVs showed reduced size and higher mass-specific energy consumption, but significantly lower whole-animal metabolic energy demand when compared with conspecifics living far from the vents [130]. It is believed that these physiological changes might allow the organisms to offset the energetic demand requested for calcification and reparation after shell dissolution occurring at the

vents. These observations also confirmed in situ transplanting experiments on benthic polychaetes, which, indeed, exhibited physiological acclimatization, metabolic adaptation, and size reduction in response to elevated $pCO_2$ in vents [132].

Calcifying organisms show different responses to acidification at SHVs. For instance, shell mineralogy, microstructure, and mechanical properties differed in four gastropod species at Vulcano SHVs [116]. Although elasticity or inner shell toughness were reduced in all four species collected from the vent area, shell length was higher in the two limpets *Patella rustica* and *P. caerulea* close to the vents compared to non-vent areas, while the opposite trend was found in *Hexaplex trunculus* and *Osilinus turbinatus*. The different response might depend on the different ability in regulating calcification rate in order to counteract dissolution under hypercapnia, an adaptation that *P. caerulea* retains even when transplanted far from the vents [133]. Another possible explanation for the observed patterns in the four species relies on the shell mineralogy, as aragonite is known to be more soluble than calcite [134] and results in more energetically demanding processes, i.e., building and replacing shell material [135]. This might be the case of *H. trunculus* and *O. turbinatus*, whose shells are entirely composed of aragonite. On the contrary *P. rustica*, living close to the vents, exhibited a significant increase in the shell calcite/aragonite ratio, similarly to what observed in the serpulid polychaete *Hydroides crucigera* and the whelk *Urosalpinx cinereal* [136]. In addition, *H. trunculus* in Vulcano SHVs displayed increased oxygen consumption rates as a result of higher energetic costs of calcification maintenance and acid–base homeostasis in the acidified habitat characterizing $CO_2$ vents (see [131] and references therein). On the other hand, the bioconstructing coral *Madracis pharensis*, which resulted highly common in sites close to Milos vents, exhibited a higher skeletal bunk density and lower porosity compared to typical constructional corals [127]. The authors proposed that the species might benefit from the input of Ca and $CO_2$ from the vents, which, together with the impulses of heat maintaining warm temperatures in winter, supported coral calcification and/or the activity of symbiotic zooxanthellae [128]. Small colonies of the endemic azooxanthellate scleractinian coral *Astroides calycularis* were found at SHVs in a cave of Ischia island [137]. Here, the colonies revealed a skeletal phenotype characterized by reduced coenosarc, encrusting morphology, fewer polyps, reduced porosity, and denser skeletons compared to colonies from non-vent sites. Along with these changes in the phenotype, the authors found a strong genetic differentiation among corals population from vent and non-vent sites. In particular, the vent population exhibited high differentiation in genes involved in calcification processes, including genes for calcium management (calmodulin and calcium-binding proteins), pH regulation (V-type proton ATPase), and carbonate localization (carbonic anhydrase) [137]. Coccolithophores assemblages, in contrast, showed disruptions in coccolith morphogenesis as a result of the exposure to low pH in Vulcano SHVs, along with decreased abundance and diversity [138].

Benthic foraminifera assemblages at the Zannone SHVs (Pontine Island, Central Tyrrhenian Sea) also showed major differences between vent and non-vent sites [139]. The exclusive presence of agglutinating species showing reduced shell size indicate the vent site as a stressful environment where chemical and physical conditions do not favor carbonate shell formation and/or preservation [139,140].

Calcifying epiphytes of the seagrass *Posidonia oceanica* exhibited a variable distribution pattern at Ischia SHVs [141]. While coralline algae including *Hydrolithon boreale*, *H. cruciatum*, *H. farinosum*, *Pneophyllum confervicola*, *P. fragile*, and *P. zonale* are usually abundant on *P. oceanica* leaves, they were completely absent in the seagrass inhabiting the vent field at Castello Aragonese. On the contrary, calcareous bryozoans *Callopora lineata*, *Electra posidoniae*, *Microporella ciliate*, and *Tubulipora* spp. were still present on seagrass leaves in the same vents. These opposite responses likely reflect the different mineralogy of the calcified organisms, with special regard to the content of Mg, which is higher in the coralline algae compared to bryozoans, making them less resistant to the chemical dissolution occurring at $CO_2$ vents (see Martin et al. [141] and references therein).

In sea urchins, a different degree of tolerance to the vent emissions was exhibited by the two common Mediterranean species *Paracentrotus lividus* and *Arbacia lixula*, with the latter been highly abundant at vents in Vulcano island [128]. This pattern can be explained by differences in the physiology of the two species, in particular in their acid–base and ion regulatory responses. Indeed, *A. lixula* displays lower coelomatic pH as well as a stronger iono-regulatory capacity compared to *P. lividus*. These features provide *A. lixula* a greater tolerance to the acidification of the waters around the vents. On the other hand, it is worth to note that, although *P. lividus* does not thrive at vents, its skeletal mechanical properties are not affected by vent emissions [142].

Acid–base regulation at hydrothermal vents is also critical for the reproduction of fishes, as demonstrated by Cattano and colleagues, who studied embryonic physiology at Vulcano SHVs [143]. In this study, embryos of *Symphodus ocellatus* collected from the vents displayed a higher degree of tolerance to $pCO_2$ concentrations due to epigenetic transgenerational plasticity providing the offspring an efficient acid–base regulatory system. Indeed, embryos of ocellated wrasse translocated from non-vent to vent areas showed a significant increase in the oxygen consumption rate, while no sign of variation was detected in embryos reciprocally translocated. Additionally, larvae hatched in the same conditions (after the above-mentioned translocation) were significantly smaller compared to the other experimental larvae. According to the authors, such changes were related to acid–base imbalances in the organisms due to an increase of internal $pCO_2$ levels and acidification of internal fluid compartments (see Cattano et al. [143] and references therein).

The presence and distribution of macroalgae in HVs can be related to their inorganic carbon physiology, especially to the functioning of Carbon Concentrating Mechanisms (CCMs). More specifically, macroalgae which do not possess CCMs are expected to overcome the Dissolved Inorganic Carbon (DIC) limitations, therefore reaching high photosynthetic rates at $CO_2$ seeps. Alternatively, by down-regulating their energetically expensive CCMs, macroalgae living in the vents can reallocate their energy towards growth or secondary processes [144]. Indeed, macroalgae thriving at Vulcano SHVs include species such as *Dictyota* spp., *Caulerpa* spp., *Flabellia petiolate*, and *Padina pavonica*, all of which either lack or can modulate their CCMs activity [145]. Additionally, the content of Chloprophyll-a and Chlorophyll-c as well as the photosynthetic rates were higher in *P. pavonica* in Vulcano SHVs compared to specimens from non-vent areas [146]. The authors suggested that the increased photosynthetic rate might have stimulated the thallus calcification in *P. pavonica*, which thrive close to vents; nevertheless, low pH is recorded at this site (see Jhonson et al., 2012 [146] and references therein). In macroalgae at Vulcano SHVs, a decrease in $\delta^{13}$C with increasing $CO_2$ was also observed [145], similarly to what found in the seagrass *Posidonia oceanica* in Panarea after the 2002 parossistic event [147]. Indeed, the increased availability of volcanic $^{13}$C-depleted inorganic carbon resulted in much lower $\delta^{13}$C signatures in plant tissues as an effect of the higher discrimination against the heavy isotope ($^{13}$C). In addition, light limitation due to turbidity and high temperature caused by the 2002 explosive event resulted in a decrease in the rhizome elongation rate. However, as the intensity of the vent emissions decreased progressively in the following months the restoration process of the plant began, reaching the complete recovery of all the considered parameters in 8 years [60]. The response of *P. oceanica* at Panarea SHVs is opposite to observations on the same species at Ischia SHVs, where the plant can thrive due to the beneficial effects from high $CO_2$ inputs [10]. Moreover, differences in the genetic expressions related to stress response have been observed in the two populations of *P. oceanica* living at Panarea and Ischia SHVs [148].

The concentration of phenolic substances, which provide protection from grazers, were found to be reduced in the seagrass *Cymodocea nodosa* living near $CO_2$ vents in Vulcano Island [149]. In particular, proanthocyaninidins and total phenolic acids decreased by 25% and 59%, respectively, with increasing $pCO_2$ and decreasing pH levels. The mechanism behind this metabolic alteration is not clear yet, as seagrasses are generally expected to

increase their phenolic content when exposed to elevated $pCO_2$ as in the case of HVs (see Arnold et al. [149] and reference therein).

Although trace element contamination was generally low in the vent area of Vulcano, Vizzini et al. [150] cautioned about a moderate potential of adverse biological impacts from hydrothermal vent input of trace elements. Nevertheless, concentrations of As, Cd, Co, Cr, Hg, Mo, Ni, Pb, and V were below detection thresholds for the limpets *Patella caerulea*, *P. rustica*, the snail *Osilinus turbinatus*, and the whelk *Hexaplex trunculus*. On the other hand, there were measurable concentrations of Sr, Mn, and U in the shells of *P. caerulea*, *P. rustica*, and *O.turbinatus*, and similarly, Sr, Mn, U, and also Zn in the shells of *H. trunculus* [151]. The concentration of Mg in the four gastropods shells was lower than those found in the leaves and epiphythes of the seagrass *Cymodocea nodosa* in the same area (cf. Vizzini et al. [150] with McClinktock et al. [151]). The difference might be a consequence of the closer interaction of the seagrass and its epiphytes with the water–sediment interface, where dissolved metals occur in higher concentration [151]. High levels of Potential Harmful Elements (PHE) in sediments and biota were observed at Panarea after the 2002 parossistic event [31]. One month after the critical event the levels of As, Cd, Pb, and Hg were significantly higher in sediments, fish, and macroalgae from the vent compared to the control area (no HVs). Ten months later, different responses in the three matrices from the vent area were observed: in particular, levels of PHE decreased in sediments, while their concentrations were still elevated in macroalgae and fishes, which showed a species-specific response due to different bioaccumulation behavior [31].

The quality of organic matter can also be influenced by the hydrothermal emissions, as measured by Ricevuto et al. [152] at Ischia SHVs. By means of Stable Isotope Analysis (SIA), the authors found out a $^{13}$C depletion and a decreased C:N ratio corresponding to higher quality of organic matter in different food sources (macroalgae, seagrass, and epiphytes) in the vents. This difference in the isotopic composition due to hydrothermal emissions was reflected in the polychaetes species *Platynereis dumerilii, Polyophthalmus pictus*, and *Syllis prolifera*, three mesoherbivore consumers known for their tolerance to the high $pCO_2$ typical of HVs [153]. According to the authors, the $^{13}$C depletion detected in vents was probably due to an increase in carbon availability and to the exploitation of volcanic-derived dissolved inorganic carbon (DIC) present in the area [152].

The vent-associated tubeworms *Lamillibrachia anaximandri* have been found at DHVs at Marsili Seamount and Palinuro volcanic complex (Western Mediterranean) as well as in mud volcanoes of the eastern Mediterranean and are considered endemic to the Mediterranean [154,155]. Unlike their eastern conspecifics, which live at a temperature of 14 °C, Palinuro tubeworms were found to live at temperatures up to 19.4 °C [156].

### 2.2. Abundance and Distribution

The structure and distribution of the benthic communities associated with HVs are strongly influenced by the extreme physico-chemical conditions, which select more tolerant or vent-specific taxa, and by temporal and spatial instabilities of active venting, which shape the living assemblages [157]. This is particularly true and evident in DHVs systems, where vents directly control the spatial distribution of macrofauna, determining a typical concentric and vertical zonation around the emission mouths, starting from symbiotrophic forms (closest to the vents) to mixotrophic and filter-feeder taxa (when sulfide concentrations decrease) [13]. Deep-sea hydrothermal communities are usually characterized by low specific diversity and high biomass dominated by symbiotrophic and vent-obligated species, and a specific taxonomic structure that repeats within large geographic regions [13]. The macrofauna of shallow-vent assemblages, on the contrary, represents a subset of the communities of surrounding non-vent areas, composed of species able to cope with or to take advantage of the hydrothermal conditions [158,159], while symbiotrophic and vent-obligated forms are absent or rare. The degree of obligacy of vent-fauna sharply changes at the depth of approximately 200 m and is considered to be the most reliable independent criterion in separating SHVs and DHVs communities [13]. Based on this criterion,

no typical DHVs communities were recorded in Mediterranean Sea so far. Nevertheless, clusters of the chemosymbiontic sibloglinid tubeworms *Lamellibrachia* spp. have been documented in association with hot hydrothermal vents on Tyrrhenian seamounts (Marsili, Palinuro) [156,160] and Aegean submerged volcanic craters (Santorini, Kolumbo) [161], as stated above. Members of this genus are symbiotrophically sustained by sulfur-oxidizing bacteria, live in sedimented areas, and are mostly common in cold seep environments, in contrast to their well-known co-familiar *Riftia pachyptila*, which are dominant species in DHVs chemosynthetic communities [156]. However, to date, vent-specific fauna has not been found in any of the Mediterranean vent areas [14,39,140]; therefore, the communities described in the present review are all considered as SHVs.

Generally, direct effect of SH venting on the macrobiota is to exclude many of the less tolerant local species [15,17,162] and to simplify benthic communities with loss in terms of species, functional, and trophic diversity in different habitats, such as rocky reefs and seagrass meadows [75,163,164].

The most known and studied effect of $CO_2$ uptake in SHVs is the reduction in abundance and diversity of calcified species, which, as described in detail in the previous paragraph, are susceptible to skeletal dissolution in low pH conditions [39,163,165–167]. This is particularly evident in studies conducted in SHVs habitats, where a pH gradient can be recognized starting from the hydrothermal emissions, allowing to better understand the effects of slightly changes of pH on the macrobenthic communities. Among these habitats, one of the most exploited in the Mediterranean Sea is the Castello Aragonese system (Ischia Island; Foo et al. 2018 [122] and references therein). Here, as the pH decreases from ambient (8.08) to low (7.8–7.5) and extremely low (6.8) values, the diversity of species shifts from a more complex and diverse macrobenthic community, including calcareous skeletons, towards a less diverse assemblage with reduction and disappearance of the calcifiers (barnacles, mollusks, and coralline algae), and finally to a simplified communities dominated by encrusting fleshy forms and non-calcifying species [164]. However, according to Foo et al. [122], some of the calcifying species (e.g., sea urchins, serpulids, bryozoans, foraminifera, and corals) living at near future (2100) ocean acidification conditions (mean pH 7.8) at the Castello Aragonese SHV show resilience to elevated $CO_2$. The same pattern is documented by Auriemma et al. [168,169] for the macrobenthos associated with the habitat former algae *Cystoseira brachycarpa* in the Bottaro crater SHV (Panarea island). Here, the changes in the structure and composition of faunal assemblages are mainly related to the pH gradient (from 8.1 to 7.8), somehow modulated by the indirect effects of the habitat complexity provided by *C. brachycarpa*, and the decrease in diversity is related to the reduction of calcareous species (mollusks) and the increase of more tolerant and opportunist species (tanaids crustaceans and polychaetes).

Simplified macrobenthic communities are also observed associated with the hydrothermal emissions in Secca delle fumose and Zannone SHV systems [42,139]. Here, the high temperatures and the presence of sulfide, coupled with the $CO_2$ uptake, drive a drastic decrease of benthic biodiversity and a dominance of most tolerant species, such as the nassariid gastropod *Tritia cuvierii* and the sediment-dwelling polychaete *Capitella capitata*, which take advantage from the high food source consisting of the microbial mats. These species show a degree of sulfide and temperature tolerance and, thus, colonize vent vicinities in large numbers in several Mediterranean SHVs. Particularly, the cogeneric nassariid *Tritia neritea* and *Nassarius mutabilis* are among the dominant macrofaunal species at the vent off Milos in the Aegean Sea [14,155,159] and at the $CO_2$ seep in Vulcano Island in the Thyrrenian Sea [130,131].

However, according to the observations of Vizzini et al. [163] on SHV of Vulcano Island, and to Molari et al. [170,171] on Basiluzzo islet (Panarea Island), the $CO_2$ of hydrothermal origin can act not only as stressor for less tolerant species, but also as a resource for primary producers. The resulting double effect leads to lower trophic diversity and a reduced efficiency in carbon transfer along the food web. Indeed, the greater biomass of primary producers and their lower-order consumers (herbivore and detritivore), supported by

the $CO_2$ uptake, are not compensated by predation of higher-level consumers (motile carnivores), which are strongly reduced or nearly absent at the $CO_2$-enriched sites.

In Milos Island SHV, the increased bacterial and photosynthetic primary production deriving from $CO_2$ uptake and hydrothermal fluids composition led to an increase in quantity and variety of suspended organic matter available for suspension and filter-feeders and a consequent increase in epifaunal diversity in vent areas [81,172]. Higher biodiversity, due to $CO_2$ uptake but also to the physical heterogeneity of vent habitat in respect to control areas, is also observed by Esposito et al. [39] in benthic communities that colonize the chimneys of Smoking land area (Basiluzzo Islet) in the Aeolian archipelago.

All the information reported so far revealed that opportunist species, which can cope with the extreme conditions of SHVs and surrounding waters, can take advantage of increased primary production, but also of the absence of predators and/or competitors. Indeed, some taxa, which are rare in other areas of the Mediterranean Sea, seem to be well adapted to intermediate hydrothermal conditions. The ectoproct *Loxosomella pes* (previously known as *Loxosoma pes*) [173], the polychaete *Platynereis massiliensis* [102,174,175], and the amphipod *Microdeutopus sporadhi* [176]) have been observed nearby the Castello Aragonese vents (Ischia island), while dense population of the Mediterranean endemic tube-dweller amphipod *Ampelisca ledoyeri* have been reported forming extensive patches on Fe-rich crusts from 80 to 120 m depth off Panarea Island [177]. Furthermore, clusters of *Spiculosiphon oceana*, a recently discovered species of giant agglutinated foraminifera [178], have been observed associated with microbial mats in Zannone and Panarea hydrothermal field [140,179]. SHVs are also important sites for the discovery of new species. In Castello Aragonese vents, four new species have been discovered in the last years: an acoel worm (*Philactinoposthia ischiae*; [180], two fabriciid polychaetes (*Brifacia aragonensis*, *Parafabricia mazzellae*; [181], and a *Posidonia oceanica* boring isopod (*Limnoria mazzellae*; [182]). A recent revision of the polychaetes genus Amphiglena Claparède, 1864, made up by Giangrande et al. [183], revealed the existence of four new species, each of which was exclusively found to be associated with SHVs of Ischia, Vulcano, and Panarea Islands. More in detail, *Amphiglena aenariensis* is described only in association with the Castello Aragonese HS (Ischia island), *A. vulcanoensis* with Levante Bay vents (Vulcano Island), and finally *A. aeoliensis* and *A. panareensis* with Bottaro crater (Panarea Island).

Although meiobenthic fauna represents a key benthic class size in all systems, few studies addressed the response of meiofaunal assemblages at HVs in the Mediterranean Sea. Colangelo et al. [184] observed that taxa distribution in the meiofauna community at Panarea vents was strongly influenced by "intermediate disturbance" due to gas bubbling, deposition of colloidal sulfur, and the coarse grain size; copepods were found to be the dominant taxon and diversity indexes showed higher values at hydrothermal sites compared to control site (no vent activity). More recently, Baldrighi et al. [185] found that total meiofauna abundance did not vary significantly between active and inactive sites in the Gulf of Naples; moreover, nematoda at the site characterized by elevated $CO_2$ emissions (H) presented the typical features of deep-sea vents with low structural and functional diversity, high biomass, and dominance of few genera, while diversity values at the site with sulfur emissions (G) were comparable to those of the inactive site. The authors suggested site G presented a condition of "intermediate disturbance" that could maintain a high nematode diversity. Soft-bottom meiobenthic and macrobenthic assemblages at HVs in the same area were characterized by the highest small-scale heterogeneities (measure of β-diversity) [186].

As regards fish in the Mediterranean Sea, an increasing number of studies, carried out mostly in the laboratory, have shown a range of direct impacts of ocean acidification on fish eco-physiology, reproduction, and behavior due to the alteration of their acid–base balance [187,188]. In particular, Milazzo et al. [187] observed that dominant males of the ocellated wrasse (*S. ocellatus*) experienced significantly lower rates of pair spawning at elevated $CO_2$ levels in Vulcano SHVs. Moreover, indirect effects such as changes in

food availability, habitat structural complexity, or predators' abundance may be the major drivers of change in the fish community structure.

Fish assemblages associated with *Cymodocea nodosa* meadows have been investigated in an SHV at Vulcano island, showing an overall well-structured fish assemblage in terms of size classes, with no differences in species richness compared to non-vent area [189]. In this case, a higher abundance of ecologically important species (such as Sparids and Labrids) were observed under acidified conditions typical of SHVs and juvenile fish abundance close to the vents showed a lack of detrimental effects. This unexpected pattern may represent a combined response of fish mobility and enhanced food resources in the acidified site, and a 'recovery area' effect of the adjacent control site. In particular, the higher abundance of the sparids *Sarpa salpa* at the SHVs site may be due to the minor content of phenolic substances, which are deterrent to herbivory. At the same time, the seagrass leaves of *Cymodocea nodosa* long-term exposed to elevated pCO$_2$ showed a greater palatability, therefore attracting the highly mobile fish species *S. salpa* to the acidified site. Similar observations were carried out by Mirasole et al. [97] at temperate SHV located around the coast of Ischia Island: in this case, authors assessed the effects of low pH on necto-benthic fish assemblages associated with the foundation seagrass species *Posidonia oceanica* in the Mediterranean Sea. The results showed a greater *P. oceanica* habitat complexity (i.e., increased shoot density) and a greater abundance of adult (especially the herbivore *S. salpa*) and juvenile fish at the vents than reference sites, while no differences were found for species richness and composition. Again, this pattern could be explained by the combined effect of a positive response to the higher structural meadow complexity and the greater seagrasses palatability/nutritional value occurring at the vents, which may help herbivores to withstand the higher energetic cost to live under high pCO$_2$/low pH conditions. Martinez-Crego et al. [190], at the same hydrothermal sites, observed doubled predatory fish densities under acidified conditions, likely due to bottom-up benefits of high pCO$_2$ on preys.

As previously stated, high pCO$_2$/low pH conditions can stimulate the productivity of many marine photoautotrophs, including seagrasses which lack effective carbon-concentrating mechanisms [149]. Studies of high CO$_2$ communities near submerged volcanic vents reveal luxurious seagrass beds such as *Posidonia oceanica* or *Cymodocea* spp. with increased shoot densities and biomass [150] and leaves devoid of calcifying fouling organisms [142]. Near Milos and Vulcano SHVs, *Posidonia oceanica* (L.) is replaced by *Cymodocea nodosa* [14,162,191], which is a more tolerant species [192], even if unable to survive in the immediate vicinity of acidic and sulfidic vents [193]. In general, with regard to algae, the research carried out in the Mediterranean Sea have shown that calcifying species at acidified sites are significantly reduced in cover and species richness, whereas few non-calcified species become dominant [194]. In particular, populations of calcifying red algae, especially crustose coralline algae (CCA), which form calcite crystals with Mg content, decrease and even disappear and are, therefore, the species most affected by acidification [96,195]. In contrast, calcifying algae that form aragonite crystals, such as CCA species of the genus *Peyssonnelia*, brown algae (*Padina* spp.), or green algae including *Halimeda* and *Acetabularia*, are more tolerant to the decrease in pH, even if with some negative cost in terms of mineralization [167]. Linares et al. [195] at Columbretes Island SHVs observed that coralligenous outcrops and rhodolith beds, mainly characterized by a large dominance of calcifying organisms, are replaced by forests of the deep-water kelp *Laminaria rodriguezii*, which becomes dominant at depths much shallower than under normal seawater conditions. In conclusion, both SHVs and DHVs in the Mediterranean Sea can support a somewhat novel biodiversity, ecologically and evolutionarily adapted to the extreme conditions or able to opportunistically exploit empty niches, reaching abundances never observed in other areas.

## 3. Prokaryotic Communities in Hydrothermal Systems

### 3.1. Survival Strategies and Adaptation

Compared to eukaryotes, which are slower and more complex, microbes are able to provide faster adaptive responses, especially in extreme living conditions [196]. The

thermal and chemical gradients occurring at DHVs [197] led to the development of microenvironments defining microbial niches [198], strongly influencing the microbial communities' establishment and their metabolic activities. Over the years, the studies conducted on HSs have surprisingly brought to light the presence of a high biodiversity, with special regards to microbial life. Both omics and classical phenotyping approaches provide the integrative suite of tools to functionally decipher the mechanisms of microbial survival in disturbed/perturbed environments [199].

Temperature influences the distribution of the microbial taxa together with the availability of acceptors and donors of electrons [27,43,200–202]. Bacteria and Archaea members inhabiting HSs are mainly represented by mesophiles, thermophiles, hyperthermophiles, and acidophiles or alkaliphiles in relation to temperature or pH, respectively [14,203,204]. Thermophiles and hyperthermophiles, living at temperatures >60 °C and >80 °C, respectively, have been detected in volcanic hot springs and HVs for bacteria [4,27,205] and for Archea [119,120,206–208]. Their proteins maintain activity at high temperatures via prominent hydrophobic cores, increased electrostatic interactions, and enrichment in salt and disulfide bridges [209]. The thermophile-specific enzyme reverse gyrase introduces positive supercoils into DNA to maintain the double-stranded helical structure at high temperatures [210]. The membranes of thermophiles feature lipids, which often form monolayers, comprising tetra-ester (bacteria) or tetra-ether (archaea) lipids enriched in branched, saturated hydrocarbon chains [211]. These adaptations ensure thermal stability, thereby maintaining nutrient transport and a chemiosmotic gradient. The highest temperature at which microbial growth has been observed is 122 °C, a record set by a DHV archaea including *Methanopyrus kandleri* 15 and strain 121 which is closely related to *Pyrodictium* isolated from a hydrothermal vent located in the Pacific Ocean [212]. It has also been stated by Mora et al. [213] that some hyperthermophiles use flagella to move near heat sources or to escape from temperatures that could be lethal. Temperature, therefore, is confirmed to play a central role in the distribution of microorganisms along the thermal gradient and in managing survival strategies to cope with fluctuating temperatures [214].

Microbial adaptations to unfavorable pH values are exhibited mainly by regulating cytoplasmic acidity/basicity levels and through mechanisms that enables the correct protein unfolding. Acidophiles, which tolerate very low pH values (<3), are able to maintain their cytoplasm close to neutral pH using powerful proton efflux pumps and proton-deflecting membranes with reduced permeability (branched ether monolayers resembling those of thermophiles [211]). Some species have an acidified cytoplasm and use chaperones to maintain correct protein folding under acidic conditions. Acidophile proteins have an overabundance of acidic residues on the surface, which minimizes low pH destabilization. Many acidophiles have been shown to be also resistant to high concentrations of metal ions, as supported by evidence of bacterial metal tolerance in the Aeolian Arc [118]. Alkaliphiles, which tolerate high pH values (>9), are able to maintain an intracellular neutral pH by cytosolic acidification, an active mechanism that uses Na+/H+ antiporters to accumulate intracellular protons [215]. A passive mechanism found in some alkaliphilic bacteria is an acidic polymer cell wall matrix. This matrix protects the membrane by preventing the entry of hydroxide ion. Finally, halophiles have also been detected in Mediterranean HVs [118]. They are special microorganisms well adapted to high concentrations of NaCl, thanks to their ability to exclude salt from their cytoplasmic protein aggregation (salt-out strategy) [216]. To survive high salinity (up to the saturation limits, about 350 g/L NaCl), these extremophiles prevent the desiccation trough osmotic movement of water out of their cytoplasm by using two possible strategies: compatible solutes and salt in strategies. A high percentage of Euryarchaeota members affiliated to the class of halobacteria was detected at the Black Point site, and to a lower extent also in the Hot Lake Site, both at Panarea vents (Aeolian Archipelago) [3]. Moreover, Gugliandolo et al. [118] isolated and characterized, phenotypically and phylogenetically, the halophilic and thermophilic *Bacillus* (*Bacillus aeolius*) from Vulcano Island. Regarding the pressure microorganisms have been found to be able to well adapt at both low or high depths [217]. To date, microbial adaptations

to high pressures have been poorly investigated in the Mediterranean HVs, probably because the maximum depth is 1200 m (Palinuro seamount), where no investigations on the microbial community have been carried out so far. Hydrothermal geochemistry influences the composition and metabolism of microbial communities that use electron sources from fluids escaping from chimneys (i.e., hydrogen sulfide, $Fe^{2+}$, hydrogen gas, and methane) for chemiolithoautotrophic growth. The free energy available from each metabolic reaction can be estimated from the concentration of reactants and products and it is a function of the chemical composition of the hydrothermal fluid, which in turn depends on the rock's geological features [218,219]. The occurrence of heavy metals and other toxic compounds as hydrocarbons or hydrogen sulfide in the hydrothermal fluids suggested the development of special adaptation strategies in SHVs [218]. Regarding the DHVs, hydrocarbons are probably released by hydrothermal activity [220,221], but very little is known about the influence of these phenomena on microbial ecology. A study carried out by Wang et al. [221] highlighted how several well-known chemoautotrophs as sulfur oxidizers actually exhibit a mixotrophic metabolism in which they can also degrade polycyclic aromatic hydrocarbons. The ability of *Erythrobacter*, *Pusillimonas*, and SAR202 to degrade polycyclic aromatic hydrocarbons have been demonstrated in DHVs, at depth far greater than those found in Mediterranean ones (up to 1200 m) [221]. These results, together with the relatively high abundance of most of the bacteria described above, highlight the potential influence of hydrocarbons in the configuration of the vent microbial communities as well as the importance of mixotrophs in hydrothermal spring ecosystems. However, new research is needed in order to evaluate the presence of hydrocarbons in association with Mediterranean HSs.

### 3.2. Abundance and Distribution

The structure of microbial communities inhabiting HVs are influenced by geochemistry, geological setting, and fluids composition [222]. Table 2 shows the microbial diversity described in different abiotic matrices of Mediterranean hydrothermal sites. SHVs provide a quicker access than DHVs to investigate the metabolic potential and the adaptation of microbial communities to extreme environments. However, comparatively few investigations of microbial communities in SHVs have been conducted [223]. The substantial difference in the composition of the microbial communities is perceived more between the mafic zone, characterized by high temperatures and acid fluids enriched with sulfites and other metals, compared to the ultramafic zone, characterized by colder fluids consisting mainly of methane, hydrogen, and hydrocarbons [224]. Ultramafic environments are often inhabited by bacteria and archaea that metabolize methane and hydrogen, while they are much less abundant in the mafic zone, which tend to be dominated by sulfur oxidizing microorganisms [225]. The concomitance of light radiation, hydrogen sulfide, and other reduced sulfur compounds in SHVs determines the presence of anoxic phototrophs. Examples of these organisms are the class of Gammaproteobacteria, represented by Cromatiales and photosynthetic members of the phylum Cloroflexi [5,57,226], although anoxic phototrophs never represented a dominant fraction of the microbial community of SHVs.

Recently, studies conducted by Price and Giovannelli et al. [34] showed that although the hydrothermal sites are characterized by well-defined microbial metabolisms, such as those of methane, hydrogen, carbon, and sulfur, due to a known composition of the fluids, they differ in the structure of the microbial community. Sulfur-oxidizing bacteria are the most widely described members of the microbial community over the years in hydrothermal environments [15,227–231], confirming the importance of sulfur metabolism in HVs. Examples of sulfur oxidizing bacteria are members of Alphaproteobacteria, Betaproteobacteria, and Gammaproteobacteria classes, widely found in SHVs at moderate temperature, around 20–50 °C [23,43,204,223,225,229,232–235]. The first studies on microbial communities in HSs were focused on the sites of the Aeolian Arc, and were based on a phenotypic and molecular characterization approach. Gugliandolo and Maugeri [228,236] were among the first highlighting the presence of obligate and facultative sulfur-oxidizing bacteria in

the SHV sulfide-containing waters of the Aeolian Islands, and concluded that the primary productivity in shallow systems is of mixed origin, namely phototrophic and chemotrophic. Specifically, through an ecophysiological characterization, they identified three distinct chemotrophic clusters, i.e., *Thiobacilius*-like bacteria originating directly from the vent-water, *Pseudomonas*-like heterotrophs from sediments near the vents, and *Thiobacterium*-like forms from both water and sediment samples. On the other hand, phototrophs are represented by Cyanobacteria members [43,237,238], as well as anoxygenic phototrophs [7,14,226].

The culture-dependent approaches are not enough to describe and improve our knowledge about the diversity of HVs microbial communities, since 99% of marine microbes are considered unculturable [239]. With the advent of new and innovative approaches, such as the molecular methodologies and the omics techniques, increasingly precise information has been obtained on the biodiversity of prokaryotes in HSs. Thus, some contributions, such as those of Giovannelli et al. [81], who explored the bacterial diversity of SHVs in Milos Island (Greece) by Denaturing Gradient Gel Electrophoresis (DGGE), as well as Lentini et al. [226], who studied the prokaryotic diversity in an active hydrothermal site located off Panarea Island (Black Point) by employing massive parallel sequencing technique targeted on the V3 region of the 16S rRNA gene, revealed a diversity rate higher than expected.

Thanks to these approaches, an abundant class belonging to Proteobacteria phylum, namely Epsilonproteobacteria [5,43], emerged as dominant only at the interface layer between the oxic and anoxic environments, called the redoxcline layer. Epsilonproteobacteria are usually abundant in the redoxcline of both water and sediment, even if they are well adapted to conditions of high concentrations of hydrogen sulfide [111,234,235]. For this reason, they could be considered one of the most representative classes only in DHV bacterial communities [34], thanks to the ability to use hydrogen, hydrogen sulfide, or thiosulfate as an electron donor, and oxygen, nitrate, or elemental sulfur as acceptors [234,235]. According to Gugliandolo et al. [240], SHVs host microbial communities phylogenetically similar to those inhabiting the deep environments, suggesting, therefore, that the Epsilonproteobacteria's versatility allows them to be a dominant class even in the shallow environment [34].

Interestingly, differences at genus level have been evidenced within the same site in relation to the kind of samples and environmental parameters. Lentini et al. [226] reported the presence of *Rhodovulum* and *Thiohalospira* members in the sediments at high temperature from Black Point site, while *Thalassomonas* and *Sulphurimonas* were detected at low temperature. In hydrothermal fluids of the same site, the abundance of *Chlorobium*, *Acinetobacter*, *Sulphurimonas*, and *Brevundimonas* was highlighted independently from temperature. Similarly, different bacterial and archaeal abundances were retrieved in high- and low-temperature samples from the Hot Lake at Panarea Island. Here, with a community dominated by Epsilonproteobacteria in both samples, the genus *Sulphurimonas* was most represented at high temperatures, while *Arcobacter* prevailed at low temperatures. Temperature has been confirmed as a real shaping parameter, as hyperthermophilic archaeal groups (i.e., Euryarchaeota) and thermophilic (i.e., Caldiserica) and thermoresistant (i.e., Firmicutes) bacterial taxa were dominant at high temperature, while non-thermophilic Bacteroidetes, Fusobacteria, and Actinobacteria were dominant in low-temperature samples [5].

Nitrate-reducing bacteria represent a large amount of chemolythoautotrophs in both SHVs and DHVs [223,235]. This statement, however, does not mean that the number of bacterial isolates from these environments is elevated, but refers to a thermodynamic calculation and to the abundance of sequences related to Epsilonproteobacteria, which play a crucial role in the nitrate reduction cycle [241].

An interesting contribution was provided by Pathwardhan et al. [242], who with a metaproteogenomic approach identified a two-stage metabolic pathway in the maturation process of bacterial biofilm occurring at Tor Caldara vents (Italy). They detected three metagenome-assembled genomes and showed that Epsilonproteobacteria were the domi-

nant chemoautotrophic sulfide-oxidizing members in the young biofilm, while Gammaproteobacteria were more abundant in the established community. The two distinct communities form as a consequence of the exposure to different sulfide concentrations and suggest a great metabolic adaptability of bacterial communities. Similarly, phylogenetic analyses of representative gammaproteobacterial sequences detected in the biofilm community of Levante Bay vent system (Vulcano Island) revealed the active presence of *Thiothrix-*, *Thiomicrospira-*, *Thioprofundum-*, and *Candidatus Marithrix*, while active members of the Epsilonproteobacteria were mainly related to *Sulfurovum* spp. These bacteria are chemolithoautotrophs that conserve energy by sulfide-oxidation, suggesting that the oxidation of reduced sulfur species is one of the main energy-yielding processes within the biofilm community exposed to hydrothermal emissions. These observations were in line with the geochemical characteristics of the area investigated ($CO_2$ and $H_2S$ availability) and with the predicted metabolic pathways, which revealed sulfide oxidation (mainly via the Sox, Fcc, and SQR pathways) and carbon fixation (via the Calvin-Benson-Bassham cycle) to be the main metabolisms [40].

Sediments rich in $Fe^{2+}$ as a result of the exhalative activity, such as those of the Diffusive ferruginous seep (DFS) of Panarea [38], have brought to light that the coexistence of ferrous ion with high concentrations of ammonium and microaerophilic conditions leads to the dominance of another class of Proteobacteria, namely the Zetaproteobacteria. The first isolates identified back in 1996 were *Mariprofundus ferrooxydans* strains PV-1 and JV-1, obtained from samples collected at Loihi Seamount near Hawaii [243]. The cultivation of these microorganisms today represents a real challenge to demonstrate the marine microbial oxidation of Fe (II), since the cultivation and the isolation of these microorganisms is not easy [244]. *Epsilonproteobacteria* and *Zetaproteobacteria*, both living in the redoxcline layer, share the great difficulty in managing oxygen concentrations in a controlled environment and are, therefore, considered among the most difficult groups to isolate in pure culture.

While bacteria over the years have been subjected to more in-depth studies and their role in biogeochemical cycles is well known [245,246], the knowledge on Archaea is scarce and their role in marine systems is still poorly understood. It is known that in the water column, they occur ranging from 5% to 30% of the total prokaryotic abundance, and this percentage tends to increase with the depth [59,247–249]. Archaea are considered as "extreme" microorganisms, as they are dominant in extreme environments, making up a large part of the microbial community [249]. Indeed, the characterization of the microbial community of both deep and shallow hydrothermal springs have highlighted the dominance of Archaea, especially in sites where environmental conditions were more extreme [34]. The hot emission areas are inhabited by Archaea belonging to the dominant Phylum *Euryarchaeota*, followed by the Phylum *Crenarchaeota* with the *Thermoprotei* class [5,25,226, 233]. Euryachaeota are an Archaea phylum that comprises the most extreme halophiles (e.g., genera *Halobacterium* and *Haloaredivivus*), methanogens (e.g., genera *Methanococcus* and *Methanothermus*), methanotrophs (e.g., ANME-1 cluster and Methanosarcina), and thermophiles (e.g., *Thermococcus*), reported as ubiquitous microorganisms in hydrothermal vents [250] and other extreme environments [251].

An interesting contribution was provided by Manini et al. [27], who explored the prokaryotic diversity in SHVs of the Mediterranean Sea (Panarea Island) and the Pacific Ocean (North Sulawesi-Indonesia). Despite the different location and ecological features, all vents showed a decrease of benthic prokaryotic abundance with increasing distance from the vents, therefore suggesting a strong influence of the vent fluids on sediment bacterial diversity. In both cases, sediments were dominated by Bacteria, but Archaea were also proven to be an important fraction of total prokaryote abundance, accounting for 18% and 27% in Pacific and Mediterranean vents, respectively.

Until a few decades ago, studies on the microbial community were conducted following the classic molecular biology techniques (MPN, Clone libraries, and DGGE), but nowadays, sophisticated sequencing techniques (Next-Generation Sequencing) are in use for more accurate and precise analysis of these particular environments. Illumina MiSeq

technology was used in a geo-microbiological study, which investigated the composition of bacterial communities in different environments of Vulcano Island, and correlated it with the composition of the gas emissions [252]. Interestingly, the results suggested a possible correlation between the microbial community composition at the genus level with the emission profiles, in particular with occurrence of decane, argon, i-octane, and undecane. From the taxonomic point of view, five phyla have been detected, namely Proteobacteria (32.22%), Planctomycetes (16.44%), Actinobacteria (13.56%), and Firmicutes (11.93%). Kilias et al. [84] used a pyrosequencing approach to study the iron microbial-mat in the shallow-submarine arc-volcano Kolumbo (Santorini, Greece) by revealing a dominating ferrihydrite-type phases and the presence of *Nitrosopumilus maritimus*, a mesophilic Thaumarchaeota strain capable of chemoautotrophic growth on hydrothermal ammonia and $CO_2$. The study suggests that the acidic SHVs could be sustained by nitrifying Archaea, as ferrihydrite-type $Fe^{3+}$-(hydrated)-oxyhydroxides in associated low-temperature iron mats could be formed by anaerobic $Fe^{2+}$-oxidation, dependent on microbially produced nitrate.

Finally, microbial diversity of bacterial symbionts in Mediterranean HVSs has also been investigated with great interest, although still not exhaustively. Indeed, Palinuro Vestimentiferan tubeworms *Lamellibrachia anaximandri* exhibited a sulfur-oxidizing chemoautotrophic style due to Gammaproteobacteria endosymbionts capable of two carbon fixation pathways: the CBB cycle and the reductive tricarboxylic acid (rTCA) cycle [156]. Marsili tubeworms also host two distinct symbionts, both sulfur-oxidizing phylotype, in their throphosome [125]. The redundancy of metabolic capabilities in the symbionts can provide advantages at DHVs where substrate concentrations are highly variable in space and time.

**Table 2.** Microbial diversity from different abiotic matrices of Mediterranean hydrothermal vent systems.

| Site | Sample | Technique | Bacteria Taxa | Archea Taxa | Reference |
|---|---|---|---|---|---|
| Thyrrhenian Sea Tor Caldara | Sediment, biofilm | Illumina HiSeq | Gamma, Epsilon, Alfa | ND * | [242] |
| Thyrrhenian Sea Basiluzzo Islet | Sediment | 454 | Flavo, Gamma, Delta | ND * | [38,39,52] |
| Thyrrhenian Sea Phlegrean Fileds | Sediment | | Gamma, Firmicutes, Bacillus, Actinobacteria | Euryarchaeota, Thaumarchaeota | [36] |
| Aegean Sea Milos | Sediment | Clone libraries | Epsilon, Firmicutes, Bacteroidetes, Aquificae | Euryarchaeota, Crenarchaeota | [43,226,232] |
| Thyrrhenian Sea Panarea Hot/cold vents | Sediment | Illumina/454 | Epsilon (both), Delta (Hot), Alfa and Gamma (Cold) | Euryarchaeota (Hot Vent) Crenarchaeota (Cold Vent) | [5] |
| Thyrrhenian Sea Panarea Smoking land | Sediment | 454 | Alpha, Beta, Gamma, Delta, *Zetaproteobacteria* | Marine Group II (MGII), the Marine Group III (MGIII), the Marine Benthic Group E (MBGE), The Deep-Sea Hydrothermal Vent Euryarchaeotal Group 6 (*Woesearchaeota*), Marine Hydrothermal Vent Group (MHVG), *Thaumarchaeota*, Miscellaneous Crenarchaeotal Group 3 (C3) | [38,39] |
| Thyrrhenian Sea Panarea Black Point | Sediment | Illumina paired-end | Alfa, Beta, Gamma, Acidobacteria, Chlorobi | Euryarchaeota Crenarchaeota Koraarchaeota | [57,226] |
| Thyrrhenian Sea Vulcano | Sediment | Illumina MiSeq | Proteobacteria, Planctomycetes, Actinobacteria, Firmicudes, Bacteroidetes | ND * | [23,226,252] |
| Aegean Sea Kolumbo | Sediment | Illumina | Epsilon, Aquificae, Delta, Gamma, Planctomycetes, Bacteroidetes | ND * | [84] |

ND *, not detected.

## 4. Biotechnological Relevance

For biotechnological perspectives of HSs, the studies are mainly focused on microbial communities, due to their capacity to shape themselves in response to environmental conditions.

The studies carried out on both shallow and deep HSs of the Mediterranean have highlighted the potential of these areas for bio-discovery or chemi-discovery, as evidenced by various experiments that have led to the isolation of new species of microorganisms. Great part of the discoveries in this context have been obtained for the SHVs of the Aeolian Arc and carried out by a group of researchers who have greatly contributed to providing new knowledge on these sites. Six new thermophilic Bacillus species from sediment of SHV at Vulcano Island have been detected [84], among which the new species *B. vulcani* proposed by Caccamo et al. [253]. Similarly, based on chemotaxonomic and phenotypic investigations, the Bacillus strain 4-1T isolated from water samples of Vulcano Island was proposed as *Bacillus aeolius* sp. nov. [118]. Three different novel Geobacillus genomospecies have been reported by Maugeri et al. [91], who investigated both the taxonomical novelty and physiological features and analyzed the fatty acid patterns of isolates. Several new bacilli have been highlighted after isolation from water and sediment samples of Vulcano, Panarea, Stromboli, and Lipari slands [93,254,255], and for some of them the biotechnological potential was also investigated (see below). The strains APA and 1A60, isolated from water and sediment samples, exhibited a similarity below 95% with GenBank deposited strains and were proposed as new Geobacillus species. Moreover, due to its lower thermophilia and higher halophilia and gelatine hydrolyse ability, the strain SBP3 was suggested as novel species of the same genus, despite the strict relation with *G. subterraneus* [93]. The identification of new prokaryotic taxonomic groups from hydrothermal vents is of considerable importance from a biotechnological point of view, as it can allow the discovery of new enzymatic arrays and bioactive molecules with a broad spectrum of application. As mentioned in the section on prokaryotic diversity, the advent and continuous evolution of new isolation and sequencing technologies have revealed the presence of taxonomic groups that are difficult to detect with traditional culture techniques. These findings have encouraged the researchers even more, supporting the idea that there is still much to discover for hydrothermal sites.

The other contributions concerning the biotechnological relevance of HSs are focused on the possibility to discover new bioactive molecules. The survival strategies that the organisms thriving in these ecosystems have developed and their adaptive responses to critical environmental changes include the production of biomolecules useful in processes involving defensive or communication strategies. Additionally, in this specific context, most of the available studies have investigated natural compounds of microbial origin, while less efforts have been spent in isolation of biomolecules from higher organisms. To the best of our knowledge, the only paper available in this context has been provided by Eythorsdottir et al. [256], who explored the bioactive compounds, specifically 3-Alkyl Pyridine Alkaloids from Sponge specimens of SHVs in Eyjafjörður, northern Iceland. The authors also investigated the biotechnological potential of symbiotic microorganisms, reporting a promising antimicrobial activity by bacterial symbionts of anemones, ascidians, macroalgae, and one nudibranch collected at the same site.

Most of the described biomolecules have been isolated from microorganisms as a prolific resource for novel compounds, and sustainable approach for bioprospecting, which allows to overcome the problems related to the recollection and over-exploitation of macroorganisms [257]. Moreover, the possibility to dive inside the whole genomes and to search for specific genes gave a significant boost to research, addressing the limits of cultivation requirements.

Despite the recognized potential of the microbial component in the bioprospecting field, at the moment, the potential of microorganisms of HVs in the Aeolian Islands as source of biomolecules remains scarcely explored. Among several biomolecules produced by thermophilic microorganisms, a great part of the studies is focused on the discovery of new enzymes and a number of extracellular polymeric substances.

### 4.1. Extremozymes

Enzymes are interesting compounds for a wide spectrum of applications in a variety of industries, such as textiles, food, chemicals, pharmaceuticals, and biofuels [258–260]. In the last decades, they have captured much more attention as they fall within the concept of green chemistry, since they catalyze faster, safer, and less hazardous reactions, and are, thus, ideal to reduce in the future humanity's overconsumption of resources [261,262]. The concrete application of enzymes remains limited to few industrial processes, due to the narrow ranges of stability in terms of external parameters, such as temperature, pressure, pH, and use of organic solvents [263]. Numerous efforts have been spent in genetic and chemical modification techniques and immobilization strategies, aimed at increasing the enzyme stability or at decreasing denaturing effect of the reaction conditions [264–266], but there is no definitive method to improve the stability of enzymes. A valid alternative to obtain active enzymes in critical conditions is to search new biological catalysts produced by microorganisms able to live in extreme environments, thus able to thrive in harsh conditions, such as thermophiles (elevated temperatures), halophiles (salt concentration), barophiles (pressure), osmophiles (osmotic compound content), metalophiles (heavy metal content), radiophiles (radiation), acid or alkaliphiles (acidic or basic pH), xerophiles (extreme dryness), psychrophiles (extreme cold), or polyextremophiles (a combination of different extremes) ([263] and therein references).

As treated in the previous sections, Mediterranean HVs host several categories of extremophiles, among which thermophiles and hyperthermophiles certainly stand out. Microorganisms living in HSs are able to produce uncommon biological catalysts suitable to support metabolic processes at very elevated temperature conditions. Indeed, they possess thermostable protein and cell membranes resistant to high temperature denaturation and proteolysis [267], thus proving to be excellent ideal producers of thermostable enzymes, named thermozymes [268–271], really intriguing for a great number of industrial applications. This feature makes them highly interesting in the industrial field, since thermal stability enables such enzymes to remain active when chemical denaturants are present or in case of harsh process conditions. This should not be surprising, considering that currently all polymerase chain reaction (PCR) performed in research laboratories are made possible by polymerases produced by thermophilic strains (e.g., from Thermus aquaticus and Pyrococcus furiosus) [268,272,273]. However, in addition to polymerases, many other enzymes have been reported from thermophilic bacteria, such as lipases, laccases, and xylanases [274,275].

Despite HVs having been extensively proved as optimal source for isolation of thermophilic and other extremophilic bacteria, they are still underexplored for the search of new enzymes, and this is particularly true for the Mediterranean HVSs. Eighteen thermophilic bacilli, closely related to Geobacillus and Bacillus genera, isolated from water and sediment samples of the Aeolian Islands of Vulcano, Panarea, and Lipari were screened for extracellular enzymes production (Tween 20, Tween 80, tributyrin, soluble starch, xylan, dextran, carragenan, gelatine, and casein) and reported at least three enzymatic activities [255]. Similarly, a screening performed on eighty-seven thermophilic spore-forming bacteria isolated from the same sites, evidenced the totality of the strains or, for most of them, the presence of gelatin and esculin hydrolysis, phosphatase, butyrate and caprylate esterase, Tween 20 and Tween 80 lipase, casein, and tributyrin hydrolase [269]. Gelatin and starch hydrolyzation and lipolytic activity were also detected in twenty-two Bacillus and Geobacillus spp. strains isolated from SHVs of Panarea [90]. An interesting approach was proposed by Placido et al. [121], who reported the activity-based survey of the metagenomic expression library generated from the environmental DNA (eDNA) from the SHV sediment of Vulcano. The study represents a valuable contribution to the field, since the authors were able to characterize three carboxyesterases and one glycosyl hydrolase among ca. 200 proteins of biotechnological relevance. Interestingly, most of the α/β-hydrolases were thermophilic and also showed high activity in the presence of heavy metals.

What emerges from the studies carried out so far is that much of the research related to extremozymes from hydrothermal environments is focused on oceanic and deep areas [276,277]. To the best of our knowledge, we have very little and fragmented information on this class of molecules from the Mediterranean HSs. Although their presence has been demonstrated, there are no in-depth studies regarding their molecular structure or resistance to extreme conditions. Furthermore, most of the more recent studies rely on function-based metagenomics, sequence-based metagenomics, and single amplified genome approaches, while these innovative techniques have never been applied to HVSs in the Mediterranean area, with the only exception of Placido and coauthors [121].

### 4.2. Exopolysaccharides

Extracellular polymeric substances and exopolysaccharides (EPSs) are generally high molecular weight substances, in the form of capsular or slime polysaccharides attached to the cell surface or dispersed in the extracellular medium as amorphous matrix. They possess an important ecological role for the survival of microorganisms, assisting them in coping with harsh environmental conditions, such as extreme values of temperature and salinity, and nutrient limitations [278]. Several different chemical structures have been described and different efficiency in EPS production have been detected according to the bacterial affiliation.

A fairly significant part of the available data on bacteria from the hydrothermal areas of the Aeolian Arc is devoted to research into the production of EPS from extremophilic strains. As pointed out by Poli et al. [278], many HSs have been proved to be new sources of EPS producing bacteria, and a wide range of biotechnological applications have been prospected. Despite this, there are still many steps to be taken. Main producers of EPS isolated from Mediterranean HVSs are affiliated to Bacillus and Geobacillus genera and were mostly isolated from water and sediment samples. Generally, the most common approach for the discovery of new producers consists of the screening of bacterial isolates on agar plates through colorimetric methods or in liquid cultures assessed with the addition of sugars. Thus, the first studies on isolates from Aeolian HVSs reported the application of standard tests for the identification of potential EPS producers and proceeded with EPS extraction by using cold absolute ethanol [91–93,118]. In all cases, a high carbohydrate content was evidenced, as well as a lower number of proteins and nucleic acids. Spanò et al. [94] provided more complete information with a deeper study on EPS produced from the strain Bacillus licheniformis T14 isolated from SHVs of Panarea. Interestingly, an optimization approach was employed to detect the better condition for EPS production, by revealing the sucrose as most efficient carbon source (final concentration 5%) and a maximum yield of 366 mg/L in fermented culture. Deeper insights have been also provided on chemical composition of the EPS, a 1000 kDa trisaccharide unit constituted by sugars with a β-mannopyranosidic configuration, and on potential applications. A strain affiliated to the same species, named B3-15 and isolated from water of SHVs at Vulcano island, has been proved to produce a maximum amount of 165 mg/L of EPS with a tetrasaccharide repeating unit, during growth in mineral medium supplemented with glucose [92,279]. *Bacillus thermodenitrificans* strain B3-72 isolated from sea water of SHV at Vulcano island showed the production of two EPSs during growth in Marine Broth at 65 °C in aerobic conditions, with the highest concentration of 70 mg/L. A trisaccharide repeating unit constituted of sugars having a manno-pyranosidic configuration was detected for the EPS2, for which the IR spectrum suggested the absence of uronic acids and sulfate residues, and mannose and glucose resulted main constituents with a ratio of 1:0.2 [280]. Similarly, fourteen strains isolated from SHVs of Flegrean Area were screened for EPS production by showing a final amount of 60 and 50 μg/L for two strains reported as Bacillus and Thermus members by morphological, biochemical, and chemical analysis [281]. The authors reported galactose and saccharose as best sugars for stimulation of EPS production by strains 4001 and 4004, and the chemical characterization of 4001-EPS showed a 380,000 D molecular structure constituted of a repetitive unit formed by seven monosaccharides.

Despite all the interesting insights, many gaps still remain in the context of EPS producer's discovery from HVs. Indeed, to the best of our knowledge, we have no report of potential bacterial producers from deep systems of the Mediterranean area. This can be explained by the strong difficulty in finding cultivable bacterial strains, but this is not enough to justify the gap. In fact, the production of EPS by bacteria from DHSs in other areas has already been reported, as in the case of the strain *Pseudoalteromonas* sp. 721 isolated from the polychaete annelid Alvinella pompejana in the active hydrothermal vent located on the East Pacific Rise, or the cases of *Alteromonas infernus* sp. nov. and *Vibrio diabolicus* sp. nov. [282]. More recently, Delbarre-Latrat et al. [283] reported Vibrio, Alteromonas, and Pseudoalteromonas strains from DHVs as the main EPS producers. Interestingly, the authors also evidenced that the EPSs showed distinct structural features, which might be useful for targeting marine bacteria that could possibly produce glycosaminoglycans-mimetic EPS.

The polysaccharides production from Archaea has been elucidated at limited extent, and available data are few and very dated [284–289]. To the best of our knowledge, the only report about EPS production by archaea in the Mediterranean areas is by Nicolaus et al. [290], who investigated the EPS production by two thermophilic archaea belonging to the genus Sulfolobus isolated from a hot acidic spring in Agnano (Naples, Italy). The chemical characterization revealed a sulfated heteropolysaccharides mainly constituted of glucose, mannose, glucosamine, and galactose.

These findings strongly suggest that the investigations in this scientific area needs to be improved, to potentially discover new EPS producer species, and new chemical structures with as-yet unknown applications.

### 4.3. Applications

The interest in the discovery of new biomolecules is dictated by different needs that can be extended to many application heads, such as medical, pharmaceutical, environmental, and industrial. Extreme environments are not all the same, each determines specific adaptations, and therefore, the possibility of finding chemicals with peculiar properties, and thus, specific advantages. The extremozymes are attractive for modern biotechnology in the context of recycling and alternative energy, as they could be used for the bio-conversion of waste and renewable substrates ([121] and references therein). Moreover, they possess the extraordinary property to be resistant to solvents, detergents, and high pressure [291], thus resulting optimal catalysts to employ in industrial processes [292]. In the era of green biotechnology, the use of marine extremozymes can be advantageous for the production of biofuels as bioethanol, biodiesel, and biohydrogen [293]. Moreover, carboxylesterases, lipases, and cellulases have been proved as particularly attractive enzymes for a lot of chemical industries, i.e., food, laundry, and pharmaceutical [294–297]. The real challenge in enzyme research is the discovery of molecules with novel enzymatic activities and improved stability [298] and, as recently reviewed by Jin et al. [276], deep hydrothermal vents are the ideal model for this type of research.

The interest in EPS molecules from HVs is also widespread in a multiplicity of industrial applications, and it is mainly assessed in the medical and pharmaceutical fields. Indeed, the modulatory effects of EPS of marine origin on the regulatory cytokines in leukocytes and tissue cells are well known. The EPS obtained by Geobacillus thermodenitrificans has been proved for the first time as dose-dependently effective for the treatment against HSV-2 replication in human peripheral blood mononuclear cells, by supporting the immunomodulatory and antiviral effects of EPS obtained by extremophilic microbes [299]. Similar findings were reported by the same research group for the EPS produced by the Bacillus licheniformis strain, able to inhibit HSV2 replication in human peripheral blood mononuclear cells through upregulation of specific proinflammatory cytokines [300]. Interesting insights have also been obtained from the EPS produced by Bacillus licheniformis T14. Spanò et al. [94] reported numerous biological activities i.e., anti-citotoxic, antiviral, immunostimulant, and antibiofilm activity and also highlighted anti-citotoxic effect of

EPS1 against Avarol in brine shrimp test, suggesting a potential use in the development of novel drugs. The further studies performed by Caccamo et al. [301], investigated the structural changes of EPS1-T14 at increasing temperatures by monitoring them through the ATR-FTIR spectroscopic technique, starting from the importance of thermostability as a prerequisite in applications that require high temperature processes. Their findings revealed a thermal stability of up to 80 °C and identified fucose and glucose as responsible of thermal tolerance and flexibility, respectively, by extending further the potential application of the EPS to the food, cosmetic, and pharmaceutical applications, and novel biomedicine areas. The EPS1-T14, together with the EPS-B3-15 have also been indicated as suitable reducing and capping agents for the green synthesis of gold and silver nanoparticles with antimicrobial properties after temperature treatments [302]. The specific studies carried out by Caccamo and coauthors [254,301] have further elucidated the chemical and physical properties of these EPSs, providing important data for their concrete applications. Fewer studies are available on the potential application of EPSs from Mediterranean HVSs in the remediation fields, despite removal of pollutants having been reported for several EPSs produced from extremophilic bacteria from other HSs [303,304] or other kind of extreme environments [305,306].

The chemical characterization of extreme molecules is the most difficult step, but is also of fundamental importance to explain their properties and to address the best suitable application of them. The high difficulty to perform the EPS determination requires the development of new protocols for purification techniques, microscopic analysis, and sensitive spectroscopic methods [307].

## 5. Conclusions

The present review underlines the importance of marine hydrothermal vent systems as ecological and bioprospecting models, with special spotlight on the Mediterranean area. Here, we evidenced that most studies are focused on shallow systems, while more fragmented is the knowledge of the deep systems both in terms of bacterial diversity and isolation of new biologically active molecules. Regarding microbial communities, the archaeal populations are drastically less explored, both in shallow and deep systems.

To fill the existing gaps in the topic, other innovative approaches could also help. This is the case, for example, of the new generation robotics and census actions based on Citizen Science and Local Ecological Knowledge, which are suitable tools to involve not only researchers but also the common people. By complying these strategies, it will be possible to continue to investigate the seabed, and to discover shallow sites and new hydrothermal emissions that are more accessible to divers.

**Author Contributions:** Conceptualization, C.R., E.A. and T.R.; investigation, E.A., R.C., V.E. and V.S.; writing—original draft preparation, E.A., R.C., V.S., V.E. and P.C.; writing—review and editing, C.R., P.C., S.C. and T.R.; supervision, C.R. and T.R.; project administration, T.R. and F.A.; funding acquisition, T.R. All authors have read and agreed to the published version of the manuscript.

**Funding:** This research was funded by Ministero dell'Università e Ricerca. Funding PON "R&C" 2007–2013—PON03PE_00203_1 Project "Marine Hazard".

**Acknowledgments:** The authors wish to thank the ECCSEL-NatLab Italy of Panarea island for the professional support.

**Conflicts of Interest:** The authors declare no conflict of interest.

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
