# Peer review of "Ecological and Biotechnological Relevance of Mediterranean Hydrothermal Vent Systems"

_minerals, doi:10.3390/min12020251_

Round 1

Reviewer 1 Report

The Review paper entitled  Ecological and biotechnological relevance of Mediterranean hydrothermal vent systems by C. Rizzo et al. represent a sound review on HVs distribution, main features, biology and with a final interesting part concerning their biotechnological relevance (I have really appreciated  this section).

I have only few minor comments to do to the MS. Please find them below:

  • line 187: authors mentioned Figure 2. Where is this figure? I cannot find it in the paper;
  • line 235: authors said  - 'Recently, some environmental parameters (temperature, redox potential, pH, oxigen concentration) have shown different pattern (data not published) compared to those previously reported'. What do authors mean with different patterns? Please specify better in few words. 
  • In sections 2. and 3. the Authors provided a detailled review on prokaryotes (the most investigated benthic component) and macrofauna. However, some studies have been done on meiobenthic component as well that deserve to be included. E.g. 1. Appolloni et al. Diversity 2020, 12, 464; doi:10.3390/d12120464; 2. Baldrighi et al. . PeerJ 8:e9058 DOI 10.7717/peerj.9058; 3. Colangelo et al. 2001 Meiofaunal biodiversity on hydrothermal seepage off Panarea (Aeolian Island, Tyrrhenian Sea). In: Faranda FM,Guglielmo L, Spezie G , eds. Structure and Processes in Mediterranean Ecosystems. Heidelberg:
    Springer, 353–359. The meiofauna is often a negleted component, but it represents a key benthic size class in all systems. 

Reviewer 2 Report

Paragraph 1.1 "Environmental parameters" does not provide important parameters such as TDS, oxygen concentration and dissolved organic matter.

There is no figure 2 in the article. There is a link to it in the text.
